# Mechanical response design of arch seat foundation under complex traffic loading

Dong Xia[1], Guanguo Liu [1]*, Dingxin Zhang[2,3], Tong Luo[2]

**1** JSTI GROUP Co., Ltd., Nanjing, Jiangsu, China, **2** School of Civil Engineering, Chongqing University, Chongqing, China, **3** Department of Civil and Environmental Engineering, National University of Singapore, Singapore, Singapore

* liuguanguo@163.com

## Abstract

The mechanical response of the arch seat foundation of extra-long-span arch bridges under complex traffic loads has not been fully understood. To address this gap, this study develops a field-scale three-dimensional finite element model (3D FEM) of the arch seat foundation based on a large- span arch bridge project in China. Accounting for complex traffic loads-including longitudinal and transverse temperature gradients, braking forces, and wind loads-a detailed analysis was performed on the stress distribution of the foundation, load transfer of vertical and slanted piles, and stress-strain behaviors of the surrounding rock. The results show that the maximum tensile stress in the foundation is localized in the connection zone between the arch abutment cap and the arch springing. Tensile forces are most pronounced at the top of vertical piles, while the maximum bending moment occurs at the pile heads. Furthermore, plastic strain and settlement of the soil-rock mass are concentrated in the interconnected regions among the arch springing, cap, and main beam. Thus, it is critical to optimize reinforcement design at these key locations, implement long-term stress and strain monitoring, and mitigate the initiation and fatigue propagation of microcracks.

## 1. Introduction

The growing expansion of transportation infrastructure into challenging terrains, such as mountainous regions, has led to the increased adoption of large-span arch bridges, owing to their superior spanning capacity and aesthetic appeal [1]. As a critical load-bearing element, the arch seat foundation supports complex loads transmitted from the superstructure and transfers them safely to the ground [2]. Under realistic traffic conditions, the stress state of the arch seat foundation becomes highly complicated, and its mechanical response directly influences the global stability and service life of the bridge. Accurate mechanical analysis and design are therefore essential to ensure structural safety, extend service life, and reduce operational and maintenance costs [3,4].

**Data availability statement:** All relevant data are within the paper and its Supporting information files.

**Funding:** This work was supported by the Transportation Science and Technology Program Project of Fujian Provincial (Grant No. ZD202401).

**Competing interests:** The authors have declared that no competing interests exist.

Recent years have seen considerable research on arch seat foundations. In the construction domain, Qian et al. [5] introduced a stepped excavation method combined with smooth blasting to address waste removal difficulties in steep terrain. Li et al. [6], Tian et al. [7], and Li et al. [8] investigated three complex geological settings-water-rich soft rock, jointed rock slopes, and karst foundations-revealing deformation mechanisms of arch seat foundations, risks of plastic zone penetration in slopes, and the stress amplification effect in karst areas. Their proposed countermeasures, including dewatering infiltration control, bolt-grouting reinforcement, and karst concrete backfilling, helped address technical challenges in controlling foundation deformation under complex ground conditions. However, these studies mainly focused on construction strategies for specific geologies and did not compare the mechanical performance of different pile types under combined loading. In studies concerning long-term stability, Yang et al. [9] used creep tests and numerical simulations to quantify stress concentration and deformation characteristics at the contact interface of a solution-collapse gravel rock foundation, establishing evaluation criteria for long-term safety of arch seat foundations in soft rock areas. For local stress and seismic performance, Zhao [10] applied a global-local finite element approach to analyze stress control during construction but did not include temperature gradients-particularly the coupling of vertical and horizontal gradients-or train braking forces in the load combinations, leading to deviations from real service conditions. Gorini et al. [11,12] developed an efficient dynamic simulation method for arch seats, overcoming the low computational efficiency of conventional continuum models. Al-Homoud and Whitman [13] experimentally and numerically validated a seismic response model for rigid arch seats governed by tilting, providing key data for seismic design. Balla and Manandhar [14] compared traditional and finite element methods (FEM), demonstrating FEM's economic and accuracy benefits in computing foundation internal forces. However, their study focused solely on internal forces and did not correlate plastic strain localization in the soil-rock improved zone with the optimization of foundation design parameters.

Although existing research has improved understanding of construction techniques in complex geology, long-term stability in soft rock, and mechanical behavior under specific loads, several limitations persist, hindering the translation of research outcomes into engineering practice. First, the combined influence of complex traffic loads-such as temperature gradients (especially the coupling of longitudinal and transverse gradients) [15,16], braking forces [17–19], and wind loads [20,21] -has not been adequately considered. Most studies analyze individual loads in isolation or fail to integrate them into a unified coupled framework. As a result, the load-transfer mechanism in the arch seat foundation–soil–structure system deviates significantly from actual bridge behavior, making it difficult to accurately capture interactive responses under superimposed loads. Moreover, key mechanical behaviors under coupled loads remain poorly understood. In particular, systematic analysis is lacking on stress concentration patterns across different foundation components and on the internal force distribution differences between vertical and slanted piles under the same loading. This gap leaves pile type selection and anti-deformation design

without a solid scientific basis. Furthermore, few studies have adopted real engineering projects as a basis for establishing full-scale models that integrate the main beam, arch footings, arch seat foundation, and slope. The relationship between plastic concentration in the remodeled soil-rock zone and foundation design parameters is also rarely examined. Consequently, many research findings fail to offer systematic guidance for precise engineering design and long-term operation and maintenance.

Given the above issue, this study develops a full-scale three-dimensional finite element model (3D FEM) based on a large reinforced concrete arch bridge project in China, integrating the main beam, arch feet, arch seat foundation, and surrounding slope. The model comprehensively incorporates complex traffic load combinations, including dead and live loads, train-induced vibrations, braking forces, global thermal effects, coupled longitudinal and transverse temperature gradients, and wind loads, as well as the embedment condition of the arch seat within the slope. A systematic analysis is conducted on the mechanical responses of the arch seat cap, vertical and slanted piles, and surrounding soil-rock mass. The study clarifies the stress distribution, deformation characteristics, load transfer mechanisms, and load sensitivity of different components under complex traffic conditions. Based on the results, targeted design optimizations and long-term maintenance strategies are proposed, with the aim of providing scientific support for the design and preservation of bridge engineering structures.

## 2. Engineering background

### 2.1. Project overview

The arch seat foundation presented in this study is part of a major bridge project in China. The bridge has a total length of 1022.1 m, with its deck connected to mountain tunnels at both ends. The main bridge is designed as a reinforced concrete arch structure with a main span of 570 m. Both arch seat foundations incorporate a composite pile system consisting of inclined and vertical piles. The overall configuration is illustrated in Fig 1.

### 2.2. Topography and geomorphology

The bridge site is situated in a low mountain river valley terrain, characterized by a U-shaped deep-cut valley. The riverbed is approximately 310 m wide, flanked by steep mountain slopes with natural slope angles ranging from 25° to 45°. The slope height is approximately 26 m, and the ridge elevation ranges from 500 m to 900 m. The site is devoid of active faults and lies between a river compound anticline and a cross-over anticline.

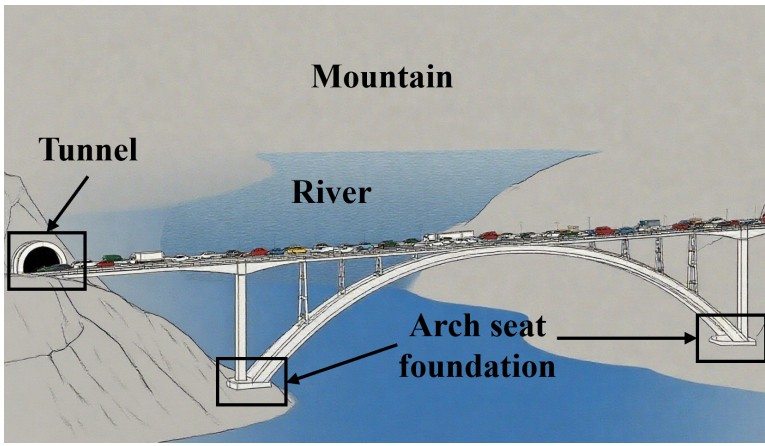

**Fig 1. Project overview schematic.**

## 2.3. Arch seat foundation

As shown in Fig 2, the arch seat foundations on both sides incorporate a composite inclined and vertical pile system, which disperses the thrust from the arch foot through a three-dimensional spatial force mechanism. The main body of the arch seat foundation is cast in C40 concrete and is reinforced with $\Phi32$ surface steel bars. Internal load-bearing steel bars ($\Phi16$ diameter) are spaced at 0.6 m × 0.6 m intervals. Vertical piles are symmetrically placed at the bottom of the arch seat cap, with a diameter of 7.4 m and a length of 34 m, mainly designed to carry vertical loads and seismic forces (Fig 2a). The slanted piles are 34 m long, with a horizontal inclination of 25°. Their cross-section is arch-shaped, measuring 6.8 m × 7.4 m, and the pile base is enlarged to 6.8 m × 8.9 m (Fig 2b). The arch seat cap, as the primary load-bearing component, is connected at its top to the box girder of the main arch and integrates vertical piles at its base. The upper right side is connected to the main arch ring at the arch foot, while slanted piles are installed on the lower left side (Fig 2c). The axes of the vertical piles align with the direction of the main beam thrust, whereas the axes of the slanted piles align with the arch foot thrust direction. This configuration enables the decomposition of horizontal and vertical forces into lateral frictional resistance and end-bearing reactions, thereby substantially reducing bending moments and shear forces on the

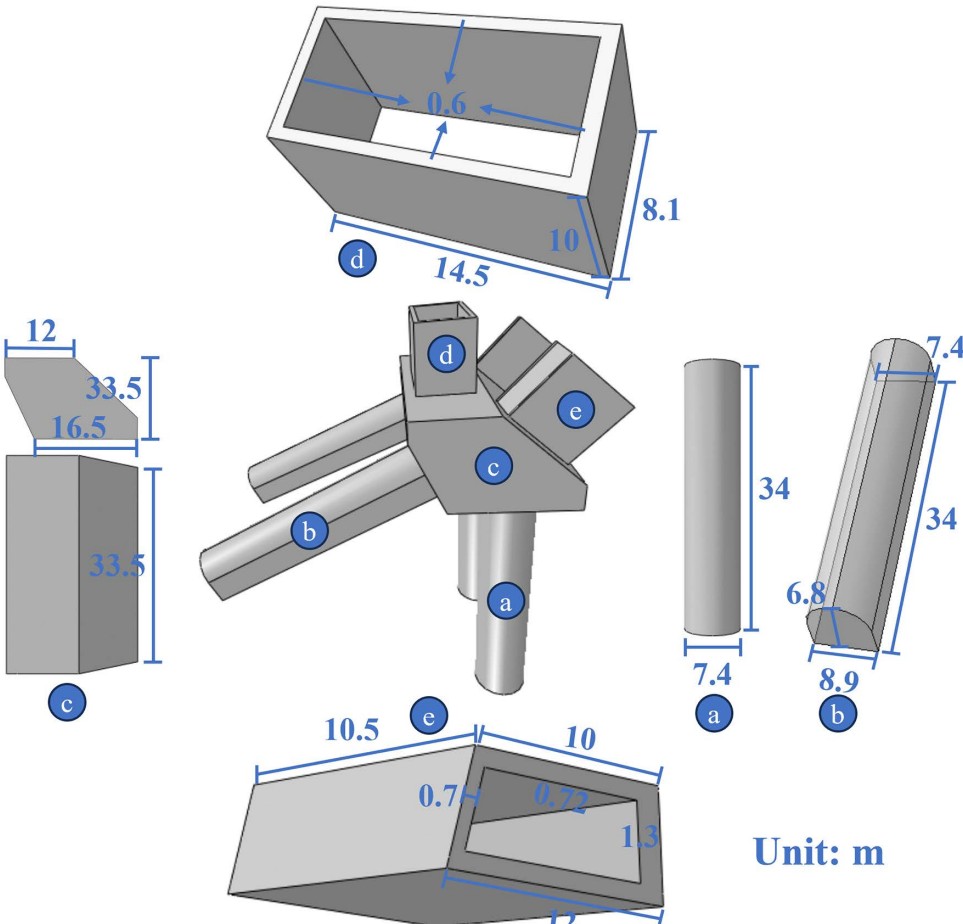

**Fig 2. Schematic diagram of the arch seat foundation structure: (a) vertical piles, (b) slanted piles, (c) arch seat cap, (d) main beam, and (e) arch foot.**

foundation. The vertical and slanted piles are rigidly connected via the arch seat cap, forming a rigid–flexible spatial force system that effectively mitigates the risk of overturning and sliding of the arch seat foundation.

The main arch girder features a single-cell box section fabricated from Q345qD steel, with dimensions of 14.5 m × 8.1 m × 10 m × 0.6 m (length × width × height × thickness) (Fig 2d). The arch foot adopts a symmetric single-cell box section made of Q420qD steel, with a trapezoidal cross-section having base lengths of 10 m and 12 m and a height of 4.6 m. The web thickness varies between 0.7 m and 1.3 m (Fig 2e).

## 3. Full-scale 3D FEM of the Arch seat foundation

### 3.1. Numerical model and boundary conditions

The finite element analysis in this study was performed using ABAQUS. As shown in Fig 3, the model comprises the arch seat foundation and the adjacent slope. Both domains are discretized with eight-node linear hexahedral elements (C3D8R) employing reduced integration and hourglass control. The geometry of the arch seat foundation follows the design parameters outlined in Section 2.3, while the slope is modeled with a length of 460 m, a width of 200 m, a height varying between 130 m and 390 m, and a slope angle of 30°. The minimum distance from vertical piles to the $X$-direction boundary is approximately 2250 m (>$30D_v$, $D_v$ represents the diameter of the vertical pile), and the minimum distance to the $Z$-direction boundary is approximately 90 m (>$12D_v$); the minimum distance from slanted piles to the $X$-direction boundary is approximately 200 m (>$24D_{s,a}$, $D_{s,a}$ represents the average width of the slanted pile), and the minimum distance to the $Z$-direction boundary is approximately 82 m (>$10D_{s,a}$). Under such geometric conditions, the influence caused by boundary effects is significantly reduced, ensuring minimal reflections and realistic stress transmission [22,23].

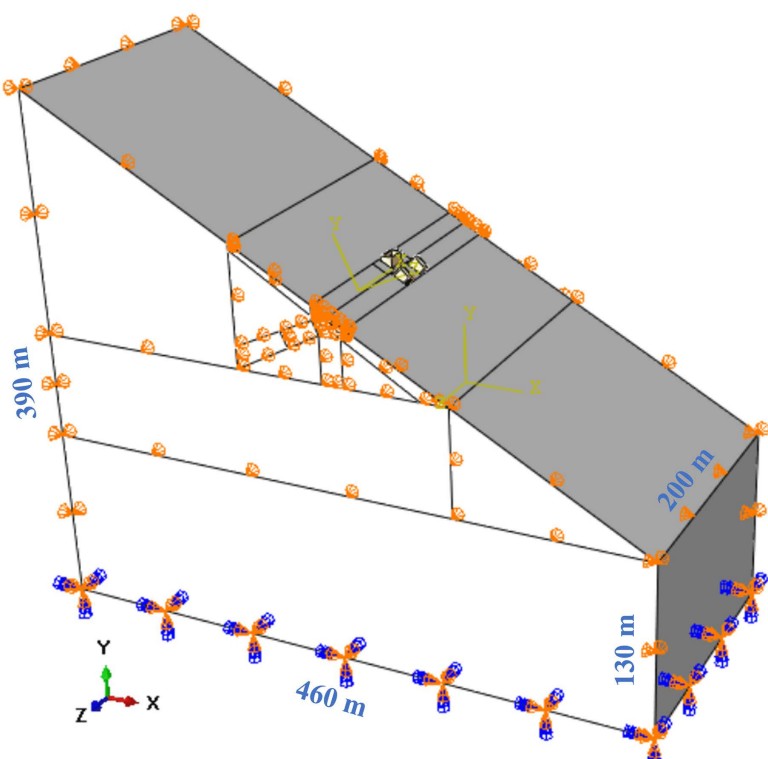

**Fig 3. Full-scale 3D FEM of the arch seat foundation.**

In accordance with real-world conditions, the vertical boundary of the slope is constrained in the normal direction ($u_z = 0$, $u_x = 0$), and fixed constraints ($u_x = u_y = u_z = 0$) are applied at the bottom boundary. General contact is employed across the entire numerical model, with a penalty friction model for tangential contact properties (friction coefficient = 0.55 [24]) and a hard contact model for normal contact.

To simulate the transmission of complex traffic loads within the cross-sections of the main beam and arch foot chamber, the upper surface of the main beam chamber was kinematically coupled to reference point R10001, and the nodes of the arch foot chamber were coupled to reference points R10002-R10009 (Fig 4). The mechanical response of the arch seat foundation was evaluated in two sequential steps: (a) Step-Geo, in which gravity was applied to the entire model and initial stresses were imported from a predefined field to achieve geostatic equilibrium; and (b) Step-Load, where concentrated forces and moments in the X, Y, and Z directions were applied to reference points R10001–R10009 to simulate the structural response under complex traffic loads.

### 3.2. Constitutive model and material parameters

In the numerical analysis presented, the extended linear Drucker-Prager (DP) model is employed to describe the stress-strain relationship of the geotechnical material of the slope surrounding the arch seat foundation. To minimize boundary effects, the geometric model is set to a relatively large size, so an elastic constitutive model (0 m-130 m) is used for soil layers at greater distances without affecting the calculation results. The yield surface F of the DP model (Fig 5) is defined by the following equation

$$F = t - p\tan\beta - d = 0 \tag{1}$$

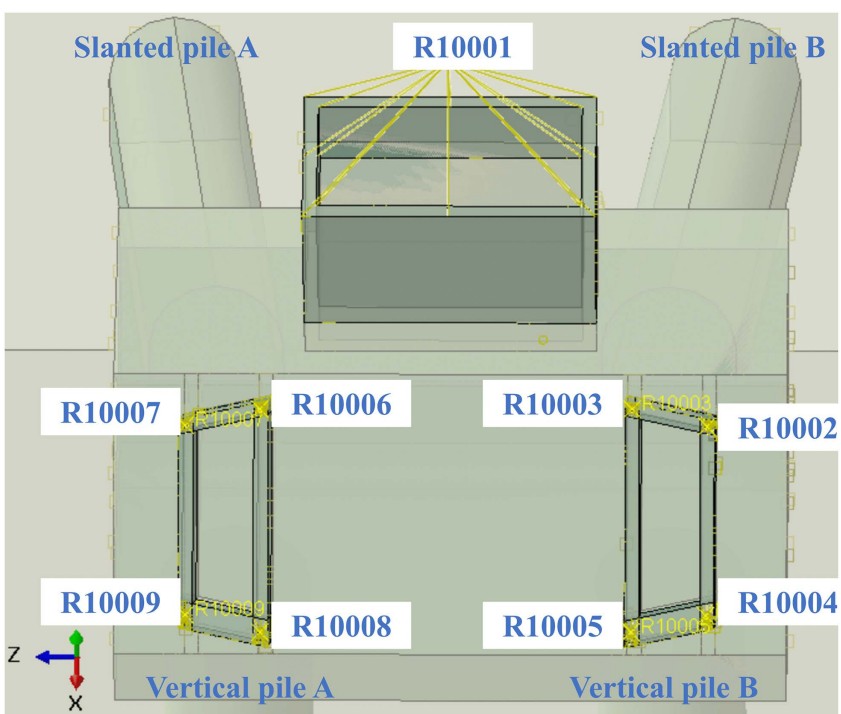

**Fig 4. Constraints of the main beam and arch foot in the numerical model.**

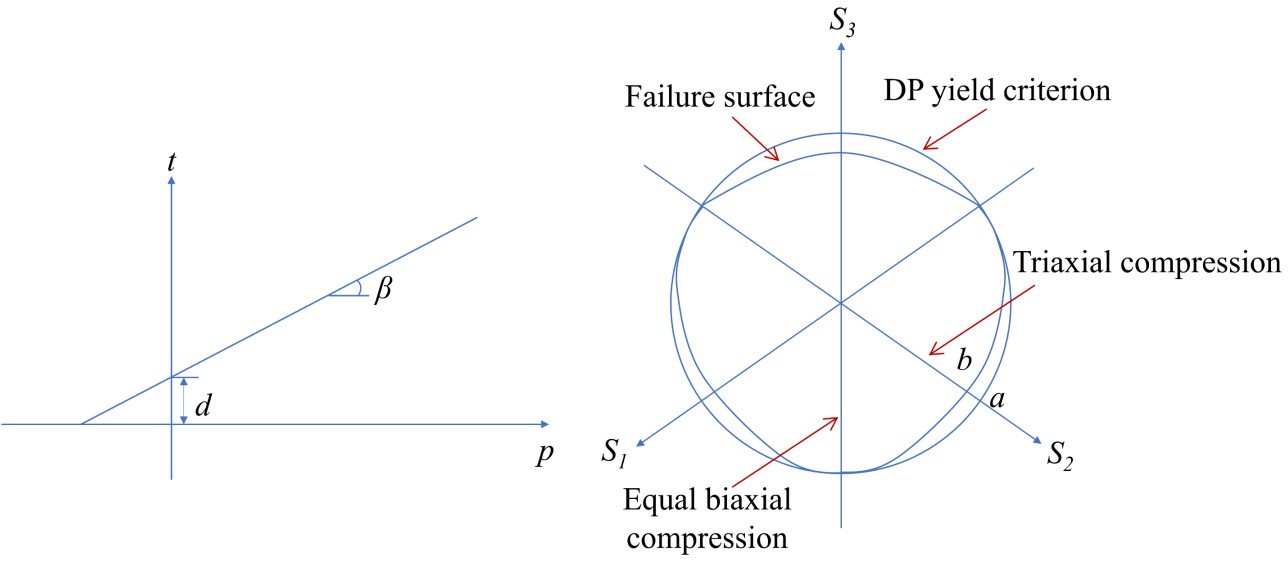

**Fig 5. Yield surface of the linear DP model [25].**

To reflect the influence of the intermediate principal stress, the yield surface in the DP model is defined by a parameter *t*, representing the deviatoric stress, and is given by

$$t = \frac{q}{2}\left[1 + \frac{1}{k} - \left(1 - \frac{1}{k}\right)\left(\frac{r}{q}\right)^3\right]$$

(2)

where $\beta$ is the inclination of the yield surface in the *p-t* stress space, related to the friction angle $\psi$, k is the ratio of the triaxial tensile strength to triaxial compressive strength, reflecting the influence of the intermediate principal stress on yielding, and *d* is the intercept of the yield surface on the *t*-axis in the *p-t* stress space, given by

$$d = \left(1 - \frac{1}{3\tan\beta}\right)\sigma_c$$

(3)

where $\sigma_c$ is the uniaxial compressive strength. The DP model uses a non-associated flow rule, with the plastic potential surface *G* (Fig 6) given by

$$G = t - p\tan\psi$$

(4)

where $\Psi$ represents the dilatancy angle. The extended DP model in ABAQUS allows for the expansion (hardening) or contraction (softening) of the yield surface, with hardening being applied in the numerical analysis presented. The variation in the yield surface size is controlled by the equivalent stress $\bar{\sigma}$, which is determined by the relationship between equivalent stress and equivalent plastic strain $\bar{\varepsilon}^{pl}$. The equivalent plastic strain is given by

$$\bar{\varepsilon}^{pl} = \int \Delta\bar{\varepsilon}^{pl}dt$$

(5)

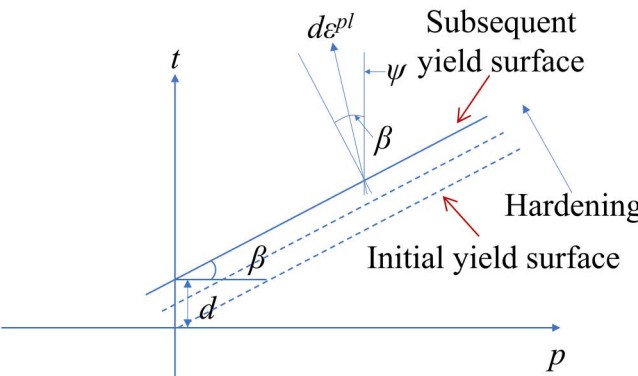

**Fig 6. Plastic potential surface of the linear DP model [25].**

For the linear DP model, when compressive hardening behavior is applied, the increment of equivalent plastic strain in ABAQUS is calculated by

$$d\bar{\varepsilon}^{pl} = \left| d\varepsilon_{11}^{pl} \right|$$

(6)

As shown in Table 1, the material parameters for the arch seat foundation, main beam, and arch foot are chosen based on the physical and mechanical properties of C40 concrete, Q345qD steel, and Q420qD steel, respectively. The material parameters for the slope are selected based on the physical and mechanical properties of the on-site main geological layer-mudstone, with a friction angle of 30° and a dilatancy angle of 15°. Fig 7 shows the comparison between the experimental results and numerical results of the uniaxial compression test of mudstone. Under unconfined pressure conditions, the axial stress of mudstone increased linearly with axial strain. When the stress reached the peak strength (i.e., uniaxial compressive strength), microcracks inside the rock initiated and propagated, and the stress decreased significantly as strain continues to increase, exhibiting obvious strain softening behavior. This phenomenon was also well captured in the numerical simulation under low confining pressure. The axial stress-strain curve of mudstone obtained from the experiment shows a good fitting effect with the numerical results, indicating that the constitutive model and geotechnical parameters selected in the numerical simulation of this paper are reasonable and applicable.

### 3.3. Mesh sensitivity analysis and numerical model validation

To evaluate mesh sensitivity, four numerical models with different element sizes were established (Table 2). The global mesh sizes-applied to the bank slope-were set to 3 m, 4 m, 5 m, and 6 m, while local mesh sizes-assigned to the abutment foundation and its vicinity-were 0.75 m, 1 m, 1.2 m, and 1.5 m, respectively. The corresponding discretized 3D FEMs are shown in Fig 8. A concentrated load was applied to the coupling reference point R10001 on the main beam cross-section,

**Table 1. Material parameters selected in numerical analysis.**

| Parts | Density (kg/m³) | Elastic modulus (GPa) | Poisson's ratio |
|---|---|---|---|
| Bank slope | 2,700 | 45 | 0.28 |
| Abutment foundation | 2,500 | 30 | 0.2 |
| Main beam | 7,850 | 210 | 0.3 |
| Arch springing | 7,860 | 200 | 0.28 |

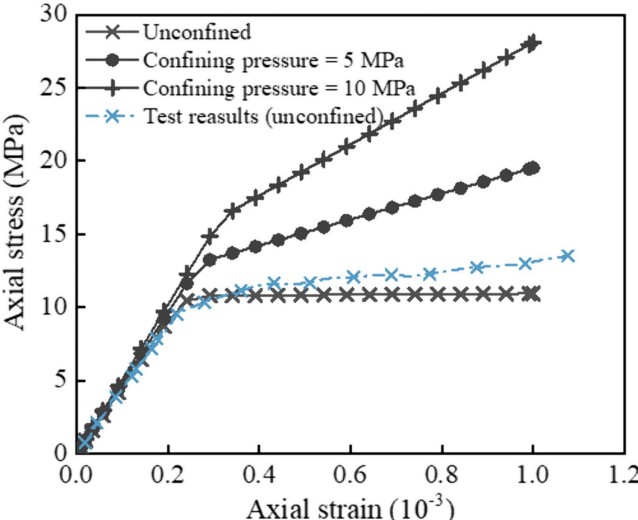

**Fig 7. Axial stress-strain curves of field mudstone samples: comparison between experiments and numerical simulations.**

**Table 2. Overview of mesh sensitivity analysis.**

| Case | Global size (m) | Local size (m) | Number of elements | Calculate costs |
|---|---|---|---|---|
| 1 | 6 | 1.5 | 162,085 | 1 hrs. 47 mins |
| 2 | 5 | 1.2 | 250,978 | 4 hrs. 12 mins |
| 3 | 4 | 1 | 459,899 | 11 hrs. 38 mins |
| 4 | 3 | 0.75 | 983,828 | 20 hrs. 19 mins |

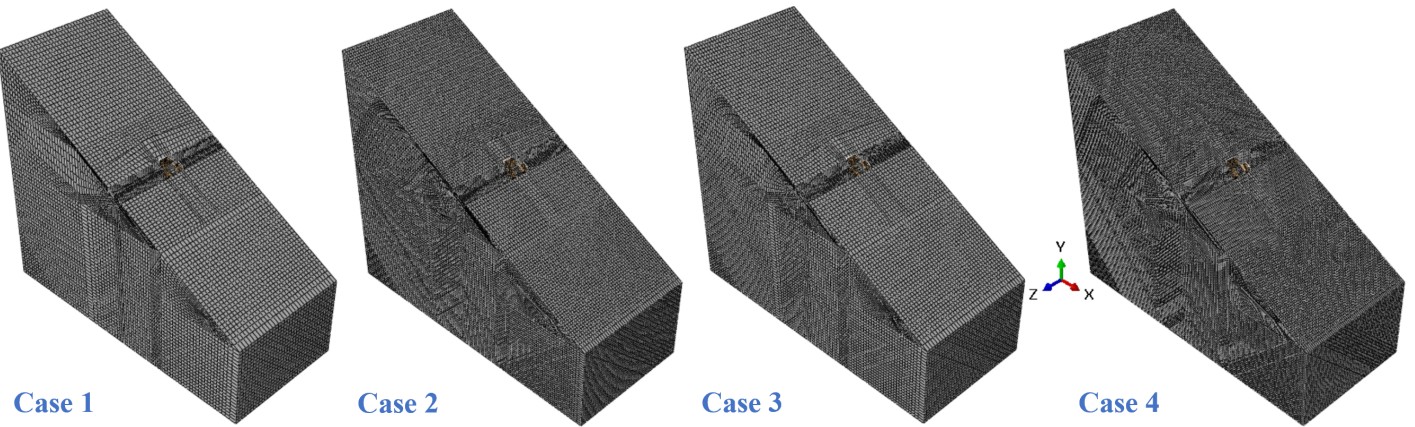

Case 1          Case 2          Case 3          Case 4

**Fig 8. 3D FEM cases with different mesh sizes.**

with components of 0 N ($X$), 128.9 MN ($Y$), and −211.9 kN ($Z$), where positive values align with the coordinate axes. The axial force distribution of vertical pile A (Fig 9) and the bending moment distribution of vertical pile B (Fig 10) were selected as representative responses. These numerical results were compared with theoretical solutions to determine a suitable mesh size for subsequent analyses and to validate the reliability of the numerical model. To minimize extraneous

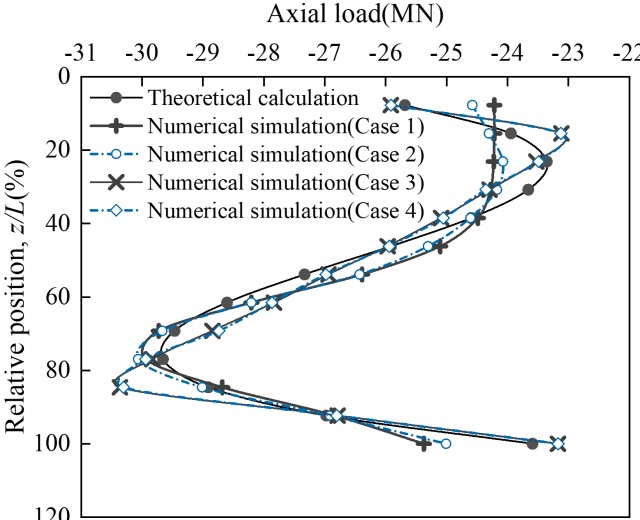

**Fig 9. Axial force distribution of vertical pile A corresponding to different mesh sizes.**

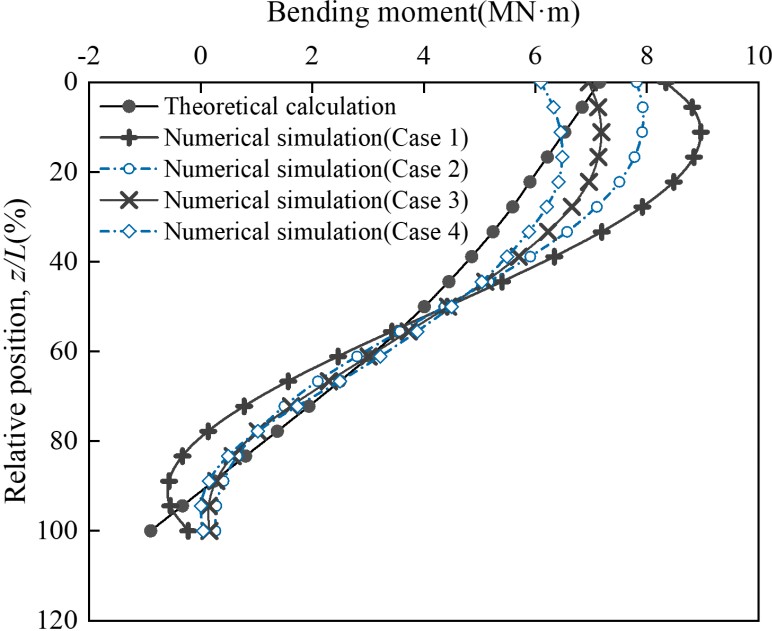

**Fig 10. Bending moment distribution of vertical pile B corresponding to different mesh sizes.**

influences, all computations from Case 1 to Case 4 were performed independently on the same high-performance computing (HPC) platform.

As illustrated in Fig 9, the coarse mesh (Case 1) did not simulate the mechanical response of the vertical pile accurately, especially near the head of the pile cap, where the deviation from the theoretical results exceeded 1.4 MN. The numerical results of Case 2 differed from those of Case 1, indicating that the results were still affected by the mesh size under this condition. The results of Case 3 showed good consistency with the theoretical calculations, validating the

feasibility of the numerical model. Although there was some deviation in capturing load transfer at 20% and 80%, the error remained within 4%. In Case 4, the mesh was further refined, but no significant change in results was observed compared to Case 3, indicating that the computational results were insensitive to mesh size under these conditions. Additionally, in terms of computational cost (Table 2), Case 4 showed an approximately 45% increase in cost compared to Case 3, while the accuracy gain was minimal. Fig 7 presents the bending moment distribution of vertical pile B under different mesh sizes. Similar to the axial force distribution, the bending moment distribution under Case 1 exhibited a significant deviation from the theoretical values, concentrated at the junction of the vertical pile and the arch abutment cap as well as the lower part of the pile. The results of Cases 2–4 achieved good agreement with the theoretical values in the lower part of the pile, while deviations still exist in the upper part of the pile. However, as the mesh density increases, this deviation gradually decreases. Through a comparative observation of Figs 9 and 10, it can be found that the mesh size of Case 3 achieves the optimal balance between "simulation accuracy" and "computational efficiency". Therefore, this mesh size is adopted for all subsequent computational analyses.

## 4. Mechanical response of arch seat foundation under complex traffic loads

Following the traffic load design and evaluation specifications for this major bridge, three main traffic load conditions (S1-S3) were considered in the numerical analysis (Table 3). The complex traffic load combinations in S1-S3 were resolved into concentrated forces and moments applied at the nodes of the main beam and arch foot, as summarized in Tables 4–6. In these tables, the most unfavorable load conditions acting on the arch seat foundation were adopted, with both concentrated forces and moments taken as the maximum values within their respective ranges.

### 4.1. Arch seat foundation stress

As shown in Fig 11, under the three complex traffic load conditions (S1-S3), a high tensile stress concentration zone appears on the inner side of the connection between the arch foot and the pile cap, with relatively smooth compressive stress on the outer side. The load transfer path is concentrated at the arch foot-pile cap connection interface. This interface serves as the "force flow hub" between the arch ring and the arch abutment foundation. Due to the significant sudden changes in stiffness and geometric structure between the arch ring box chamber and the arch seat cap, the axial force, bending moment, and shear force transmitted by the arch ring undergo load redistribution at this interface. Such stiffness discontinuity directly leads to severe concentration of stress flow inside the interface. Furthermore, the connection between the arch springing and the cap results in material deformation incompatibility (differences in elastic modulus and

**Table 3. Complex traffic loads considered in numerical analysis.**

| Number | Loading Condition | Load combination description |
|---|---|---|
| S1 | Main force and longitudinal additional force | Main force (constant load[a] + moving load[b] + train sway force) + overall temperature effect + vertical temperature gradient effect + train braking force[c] |
| S2 | Main force and lateral additional force under traffic conditions | Main force + overall temperature effect[d] + horizontal temperature gradient effect + wind load (with trains)[e] |
| S3 | Main force and lateral additional force under no-train conditions | Main force + overall temperature effect + horizontal temperature gradient effect + wind load (without trains) |

The load conditions are specified as follows:

a. Primary dead load, secondary dead load, concrete shrinkage and creep, and uneven settlement of the arch seat foundation;

b. Single-track ZKH live load and lateral sway forces;

c. Vertical static live load considered at 10%;

d. A global temperature increase of 14°C and decrease of 19°C, considering vertical and horizontal temperature differentials;

e. Wind load intensity calculated according to the "Code for Design of Railway Bridge and Culvert" (TB 10002−2017) [26], with a basic wind speed of 25 m/s.

**Table 4. Concentrated force and bending moment at each node in S1.**

| Node | $F_X$ (kN) | $F_Y$ (kN) | $F_Z$ (kN) | $M_X$ (kN·m) | $M_Y$ (kN·m) | $M_Z$ (kN·m) |
|---|---|---|---|---|---|---|
| R10001 | −1,140.0 | 138,461.7 | −212.1 | 23,876.8 | 7,521.6 | 60,401.2 |
| R10002 | 48,549.7 | 47,245.6 | −10,740.6 | 3,257.7 | 3,655.5 | −10,639.0 |
| R10003 | 56,047.6 | 51,806.4 | −6,462.1 | 365.8 | 2,745.9 | −10,930.6 |
| R10004 | 85,142.0 | 73,119.5 | −5,620.1 | 2,850.2 | 5,243.6 | −21,323.9 |
| R10005 | 94,359.1 | 87,170.1 | −6,941.1 | −650.3 | 1,646.5 | −22,004.8 |
| R10006 | 55,827.1 | 51,520.4 | −12,749.8 | 1,895.4 | 1,474.9 | −13,919.4 |
| R10007 | 48,199.8 | 46,994.7 | −11,670.7 | 6,824.5 | 2,548.8 | −13,414.6 |
| R10008 | 94,424.5 | 87,564.1 | −12,934.8 | 1,907.9 | −422.3 | −21,979.2 |
| R10009 | 84,913.7 | 73,224.3 | −15,673.3 | 3,579.3 | 3,391.2 | −21,283.3 |

$F_X$ represents the reaction component in the X-axis in the global coordinate system.

$M_X$ represents the moment component about the X-axis in the global coordinate system.

**Table 5. Concentrated force and bending moment at each node in S2.**

| Node | $F_X$ (kN) | $F_Y$ (kN) | $F_Z$ (kN) | $M_X$ (kN·m) | $M_Y$ (kN·m) | $M_Z$ (kN·m) |
|---|---|---|---|---|---|---|
| R10001 | 0.0 | 138,252.5 | 1,358.4 | 143,933.4 | −3,403.4 | −29,511.0 |
| R10002 | 60,332.1 | 56,832.5 | −9,538.2 | 3,614.4 | 3,401.6 | −12,605.4 |
| R10003 | 60,189.7 | 55,647.1 | −6,740.2 | 160.2 | 1,319.2 | −12,594.4 |
| R10004 | 91,414.8 | 78,299.3 | −4,698.0 | 2,899.0 | 5,308.0 | −21,835.8 |
| R10005 | 89,889.8 | 81,887.3 | −6,605.2 | −339.7 | 1,473.5 | −22,431.5 |
| R10006 | 55,202.1 | 50,775.7 | −13,012.5 | 1,067.3 | −445.9 | −9,841.3 |
| R10007 | 39,017.9 | 39,614.4 | −10,797.0 | 6,383.3 | 2,153.1 | −9,220.4 |
| R10008 | 96,342.5 | 89,200.3 | −12,600.0 | 1,979.0 | −412.4 | −21,924.4 |
| R10009 | 76,283.9 | 64,914.8 | −14,434.0 | 3,976.5 | 3,456.4 | −21,149.5 |

**Table 6. Concentrated force and bending moment at each node in S3.**

| Node | $F_X$ (kN) | $F_Y$ (kN) | $F_Z$ (kN) | $M_X$ (kN·m) | $M_Y$ (kN·m) | $M_Z$ (kN·m) |
|---|---|---|---|---|---|---|
| R10001 | 0.0 | 129,123.8 | 2,089.9 | 212,053.0 | −10,159.3 | 6,419.9 |
| R10002 | 52,252.1 | 48,681.9 | −8,313.8 | 3,132.9 | 3,071.1 | −10,881.8 |
| R10003 | 46,800.3 | 43,498.3 | −6,851.6 | 4.7 | 923.8 | −10,651.4 |
| R10004 | 86,433.0 | 71,984.0 | −3,136.2 | 2,733.0 | 4,772.8 | −19,873.8 |
| R10005 | 79,839.2 | 70,120.0 | −6,483.5 | −514.9 | 1,318.3 | −20,348.2 |
| R10006 | 38,667.1 | 35,739.5 | −13,096.1 | 1,006.1 | −1,054.6 | −6,625.7 |
| R10007 | 21,835.7 | 24,039.9 | −8,827.6 | 6,247.7 | 1,656.6 | −5,997.7 |
| R10008 | 87,900.9 | 79,505.3 | −12,434.3 | 1,609.8 | −492.4 | −19,530.7 |
| R10009 | 64,936.4 | 52,929.5 | −12,903.5 | 3,730.3 | 3,108.0 | −18,822.9 |

Poisson's ratio), and the curved surface of the arch springing generates radial tensile stress components. Both of these factors collectively induce the superposition of tensile stresses inside the interface. Under condition S2, the peak tensile stress reaches its highest value (23.70 MPa). This results from the combined effect of the additional bending moment induced by the transverse temperature gradient and the horizontal wind load during vehicle presence, leading to a steep stress gradient. In condition S1, the peak tensile stress is 21.38 MPa, accompanied by a relatively minor additional bending moment due to vertical temperature gradients, along with longitudinal braking forces. These contribute to a narrower

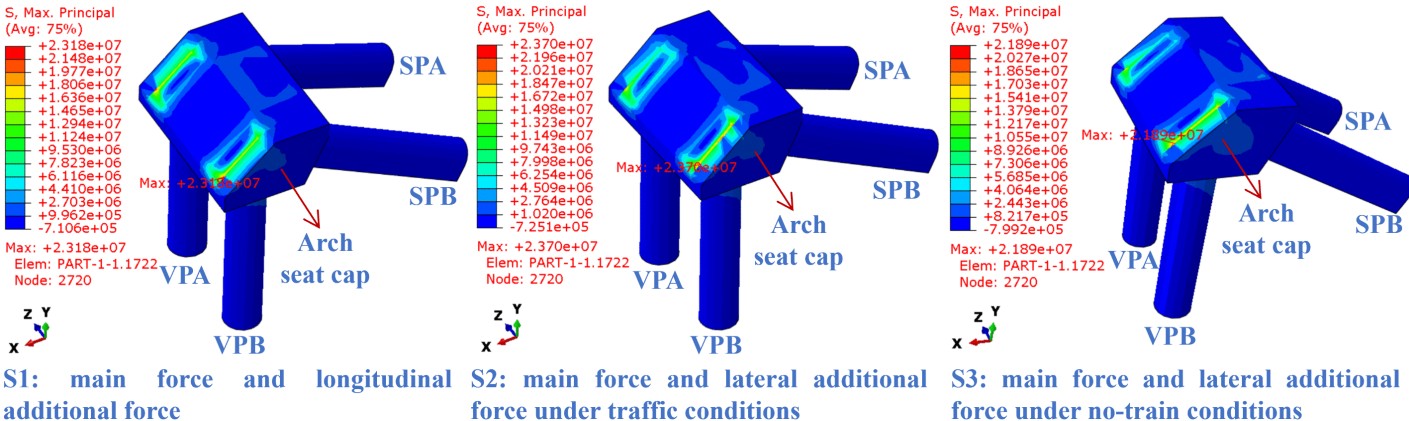

**Fig 11. Maximum principal stress contour plot for the arch seat foundation.**

stress concentration zone. Under condition S3, the peak tensile stress is 21.89 MPa. In the absence of vehicle loads, the stress distribution is more uniform. Among the three load combinations, the maximum compressive stress occurs under condition S3 (−7.99 MPa). The presence of compressive stress on the outer side of the foundation suggests that wind load exerts a more pronounced lateral compression effect when no vehicle load is applied. Under this scenario, the vertical compressive stress contribution from the primary load decreases, thereby increasing the proportion of horizontal compressive stress. The compressive stress values decrease sequentially in conditions S2 (−7.25 MPa) and S1 (−0.71 MPa). In condition S1, braking forces acting longitudinally contribute minimally to horizontal compressive stress at the foundation. In addition, vertical temperature gradients enhance the tensile tendency of the foundation, partially offsetting the compressive stress and resulting in a lower absolute value of compressive stress. Such cyclic and repeated action of high tensile stress tends to induce the initiation and fatigue propagation of microcracks under repeated traffic loads. In the long term, this may reduce the effective force transfer performance of the interface, posing a potential threat to the overall durability of the arch bridge. To mitigate this, engineering measures such as optimizing the reinforcement detailing at this interface (e.g., increasing bar diameter and density based on the stress concentration magnitude observed in Fig 11) and deploying long-term strain monitoring systems to track stress evolution are crucial.

As shown in Fig 12, under the three traffic load conditions (S1-S3), tensile stress is concentrated at the connection between the slanted piles and the foundation, while compressive stress localizes at the pile toes; no scattered high-stress zones are observed. In condition S1, the maximum tensile stress at the left slanted pile SPA reaches 0.93 MPa. This is attributed to longitudinal deformation discrepancy between the pile cap and the slanted piles induced by the vertical temperature gradient, combined with the longitudinal braking force and primary load, which together lead to a significant concentration of longitudinal force at SPA and a pronounced stress gradient. Under condition S2, the maximum tensile stress at SPA is 0.89 MPa, with a broader stress concentration zone resulting from transverse loading, though the peak value is slightly lower than in S1. In condition S3, the right slanted pile SPB exhibits a maximum tensile stress of 0.79 MPa. Absence of vehicle load reduces the load combination effect, resulting in a wider tensile stress concentration zone and a more gradual stress gradient. The slanted piles are constructed of C40 concrete, which has a characteristic axial tensile strength of approximately 2.39 MPa. In all cases, the peak tensile stress remains well below this limit, indicating that the pile body operates within the elastic range. Additionally, the embedded reinforcement helps optimize stress distribution and improves structural ductility. However, the cyclic stress action induced by repeated traffic loads may still induce fatigue damage. Especially in stress concentration zones, the steel-concrete interfaces are prone to the accumulation of fatigue cracks. Although the current stress level has not reached the ultimate strength, the long-term fatigue effects

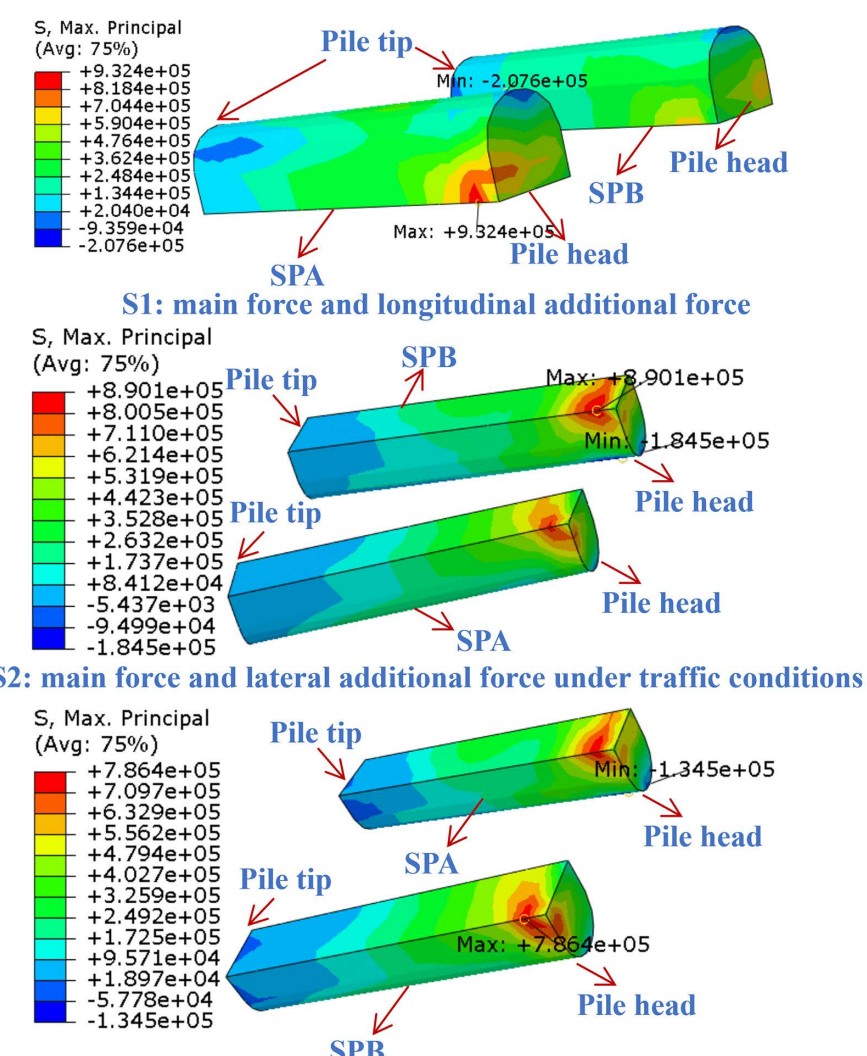

**Fig 12. Maximum principal stress contour plot for the slanted piles.**

should be incorporated into the structural full-life assessment. Based on the stress distribution (e.g., the region of rapid stress gradient change in SPA), refining the reinforcement layout at the slanted pile-foundation connection (e.g., adopting lap splices or additional stirrups) and implementing periodic crack width monitoring, which directly responds to the fatigue risk indicated by the stress cycling characteristics.

Fig 13 presents the maximum principal stress contours of the vertical piles. Under all three conditions, the principal stress distribution exhibits a pronounced concentration at the pile head, with the stress gradient diminishing along the pile shaft. In condition S1, the maximum principal stress at the left vertical pile (VPA) reaches 2.74 MPa. The pile head region, subject to constraint from longitudinal deformation induced by the vertical temperature gradient, forms the core stress concentration zone. Under condition S2, the maximum principal stress is approximately 2.57 MPa. The inclusion of transverse temperature gradients and vehicular wind load causes the stress concentration zone to extend toward the upper-middle section of the pile. Although the peak stress is slightly lower than in S1, the coupled loading effect leads to

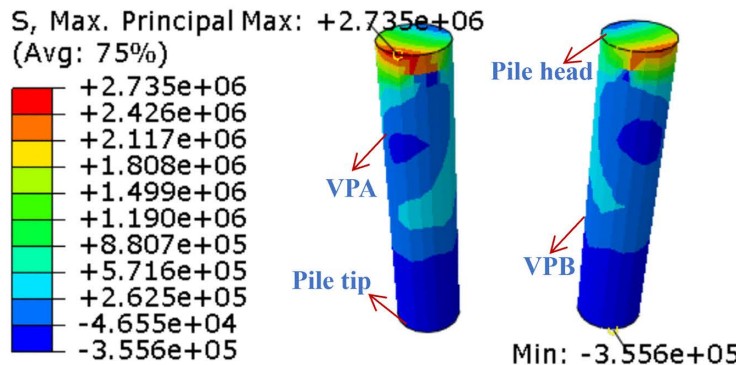

S1: main force and longitudinal additional force

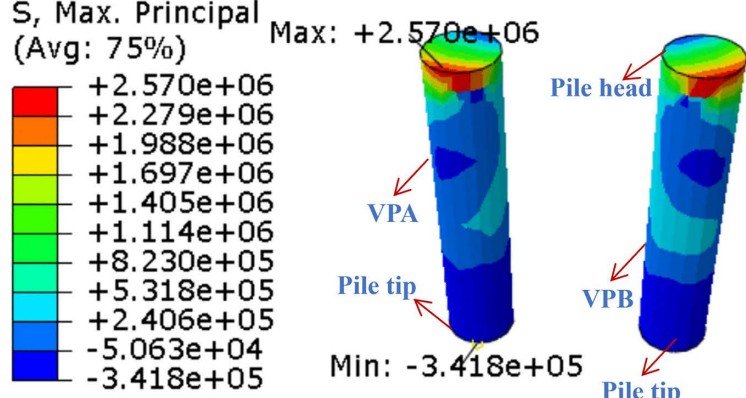

S2: main force and lateral additional force under traffic conditions

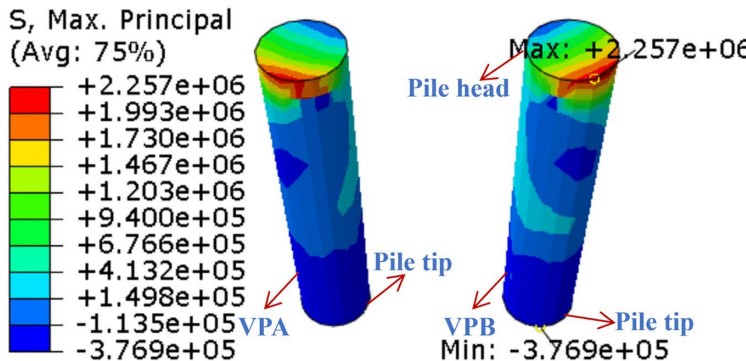

S3: main force and lateral additional force under no-train conditions

**Fig 13. Maximum principal stress contour plot for the vertical piles.**

more pronounced disruption in the lateral stress distribution along the pile. In condition S3, the maximum principal stress decreases to 2.26 MPa. In the absence of vehicle load, the stress concentration zone becomes more diffuse, though the pile head remains a stress-critical region. Under repeated traffic loads, cyclic stress concentration at the pile head, as quantified by the peak values (up to 2.74 MPa in S1), tends to induce fatigue cracking. Gradual propagation of micro-racks weakens the effective cross-sectional stiffness of the pile shaft, while the stress redistribution of steel reinforcement

and crack inhibition effect are particularly critical in delaying the fatigue damage process. Specifically, engineering practices should focus on enhancing the pile head reinforcement (e.g., increasing longitudinal bar diameter and adding confinement reinforcement) and deploying stress-strain gauges at the pile head, as the numerical findings (stress gradient and concentration magnitude) directly justify these measures for long-term fatigue resistance. This also highlights the necessity of long-term monitoring and fatigue performance optimization for the pile head region.

## 4.2. Axial force distribution of vertical and slanted piles

Fig 14a illustrates the axial force distributions of vertical and slanted piles under conditions S1–S3, revealing significant differences in mechanical behavior. In the vertical piles, tensile forces develop near the pile head and transition linearly into compressive forces toward the pile tip (tensile forces are defined as positive). The peak tensile force at the pile head decreases sequentially from S1 to S3, whereas the peak compressive force at the pile tip increases. Under condition S2, the maximum tensile force at the head of vertical pile VPB reaches approximately 24.24 MN. In condition S3, the maximum compressive force at the tip of vertical pile VPA attains about 26.93 MN.

As shown in Fig 14b, in contrast to the vertical piles, the slanted piles develop compressive forces near the pile head. The compressive force decreases along the pile shaft toward the tip, resulting in an overall compressive state along the pile. The maximum compressive force occurs at the head of slanted pile SPA under condition S1, with a value of approximately 16.97 MN. Owing to their distinct layout configurations, the vertical and slanted piles exhibit differentiated load-resisting mechanisms: the vertical piles show "tensile force concentration at the pile head", while the slanted piles exhibit "compressive force concentration at the pile head". Moreover, the extreme values of axial forces are sensitive to variations in load combinations.

The influence of traffic load combinations on the consistency of axial force distribution also differs between the left and right sides of vertical and slanted piles (VPA and VPB, SPA and SPB). Under condition S1, the axial force distributions of vertical piles VPA and VPB are nearly identical. However, with the introduction of wind load in conditions S2 and S3, the load transfer behavior begins to diverge, with VPA exhibiting more pronounced tensile action. In condition S3, the discrepancy in axial force distribution between VPA and VPB becomes most significant, showing a difference of approximately 8

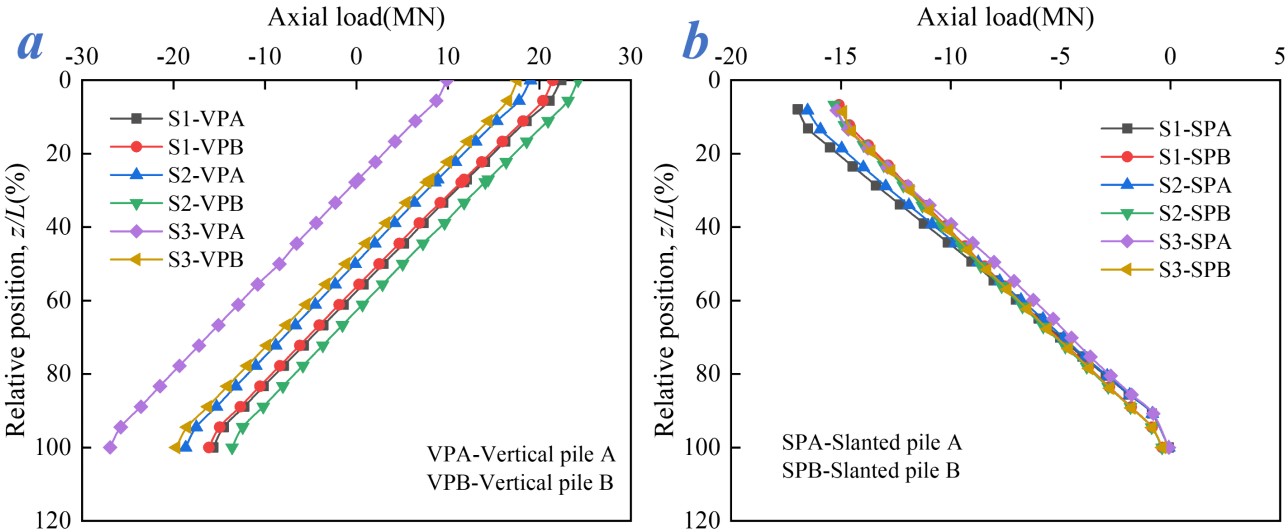

**Fig 14. Axial force distribution: (a) Vertical pile, (b) Slanted pile.**

MN at various cross-sections. This indicates that transverse wind load disrupts the symmetry of load transfer in vertical piles.

In contrast, the axial force distributions of the slanted piles remain highly consistent across conditions S1-S3, with only minor deviations near the pile head. This reflects the stronger resistance of the inclined configuration to transverse load disturbances. The ability of the slanted piles to convert transverse wind load into axial pressure along the pile axis helps maintain uniform stress distribution, effectively balancing deviations induced by asymmetric loading.

Under condition S1, the vertical load induces tensile forces at the head of the vertical piles, while the slanted piles develop compressive forces at the head due to longitudinal horizontal thrust. In condition S2, the introduction of wind load (with vehicle) disturbs the force equilibrium between the left and right vertical piles, with the wind load significantly affecting VPB and leading to the maximum tensile force at the pile head. Under condition S3, the absence of vehicle load diminishes the primary load effect, and the cumulative load transfer at the tip of VPA results in the maximum compressive force, thereby amplifying the force disparity between the left and right sides of the vertical pile.

Hence, monitoring should be prioritized at the tip of VPA (exhibiting maximum compression in S3) and the head of VPB (experiencing maximum tension in S2). Regular stress assessments are recommended to prevent material degradation under long-term loading. Meanwhile, the slanted piles, benefiting from their slanted configuration, effectively convert transverse wind loads into axial compression along the pile shaft, thereby maintaining favorable stress uniformity. Nevertheless, attention should still be given to the local stresses at the head of SPA (zone of maximum compression in S1).

## 4.3. Bending moment distribution of vertical and slanted piles

Fig 15a shows that the bending moments of vertical piles VPA and VPB are distributed symmetrically along the pile shaft, owing to the rigid connection at the pile head with the arch seat cap and the rock-slope anchoring constraint at the pile tip. The bending moment increases toward the pile head and diminishes to nearly zero at the tip. This distribution pattern remains largely consistent across different load combinations, with the maximum bending moment (approximately 40 MN·m) occurring at the pile head. Primary loads and thermal effects are transmitted to the pile head via the superstructure, while the rock slope restrains tip rotation, leading to bending moment concentration at the top.

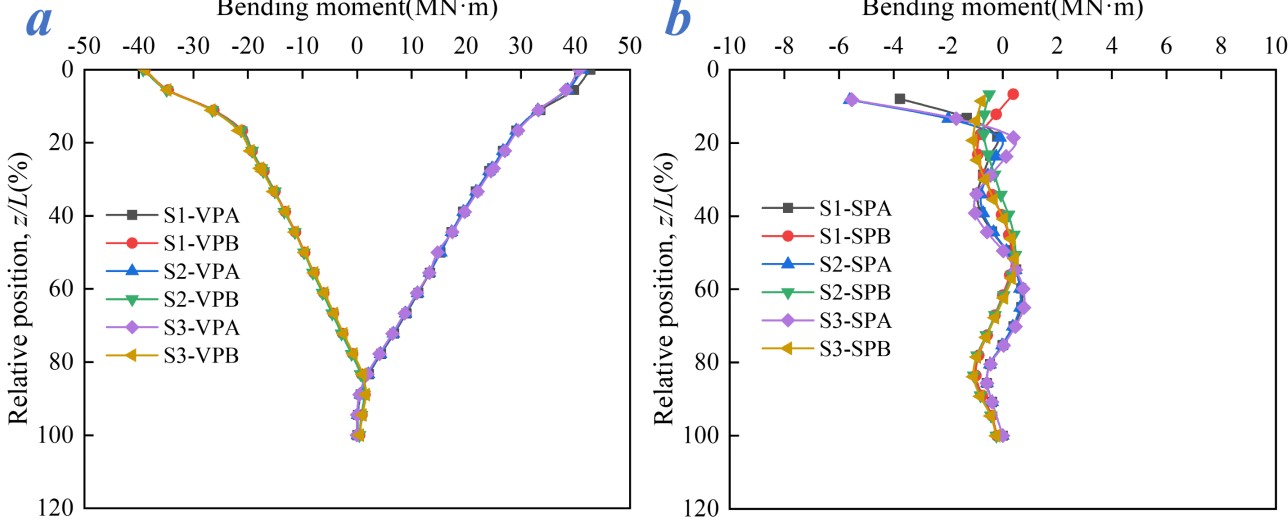

**Fig 15. Bending moment distribution: (a) Vertical pile, (b) Slanted pile.**

In contrast, due to their inclined arrangement, the slanted piles experience stronger horizontal restraint at the head from the arch seat foundation, together with increasing resistance from the surrounding soil along the shaft. The bending moment follows a "high-at-head, decaying-to-zero" pattern along the pile, with a maximum value of about 5.8 MN·m. This distribution is only marginally influenced by load combinations. Owing to the uniform rock slope constraint and symmetrical inclination, the bending moment distributions of slanted piles SPA and SPB also exhibit considerable symmetry. The differences in bending behavior stem primarily from the pile layout: vertical piles are installed vertically, with rock-slope constraints concentrated at the tip and vertical load transfer dominating, leading to bending moment accumulation at the head. Slanted piles, by contrast, are installed at an angle. Their bending moment results from a combination of horizontal load components and axial soil resistance, and thus decays along the shaft.

Under conditions S1-S3, the braking force (S1) intensifies the bending moment at the head of vertical piles, while wind loads (S2, S3) more significantly affect the slanted piles due to their transverse action. Nevertheless, the stabilizing effect of the rock slope results in only minor overall variations in bending moment distribution. Therefore, during operation, it is essential to monitor bending moment concentration at the heads of both vertical and slanted piles and to correlate these with rock slope deformation to prevent cracking. The anchoring characteristics of the rock slope should be utilized to enhance reinforcement at the tips of vertical piles and to improve the geotechnical anchorage in the inclined portions of slanted piles, thereby strengthening their bending resistance.

## 4.4. Equivalent plastic strain in the soil and rock mass surrounding the arch seat foundation

As shown in Fig 16, under conditions S1-S3, the plastic strain in the soil-rock mass surrounding the arch seat foundation is highly concentrated in the right-side arch foot–foundation connection area, with negligible plastic deformation occurring elsewhere. As a critical load-transfer interface, the arch foot transmits primary loads, thermal stresses, braking forces, and wind loads from the superstructure, thereby inducing high stress concentrations in this region-a pattern also observed in Fig 11. The rock slope imposes strong anchoring constraints along the pile shafts and beneath the pile cap, which restricts the development of plastic strain in the surrounding soil and deeper slope regions. Consequently, plastic deformation is confined to the "rigidity transition zone" between the arch foot and the pile cap, where stress concentrations exceed the elastic limit of the geomaterials, giving rise to localized plastic zones. It is thus recommended to strengthen the soil-rock mass in this connection area to raise its elastic limit and mitigate plastic deformation. Fig 17 presents the vertical settlement of the soil-rock mass around the arch seat foundation. Under all three load conditions, settlement exhibits a load-driven pattern, concentrated mainly in the connection zone between the arch foot, foundation, and main beam. Loads

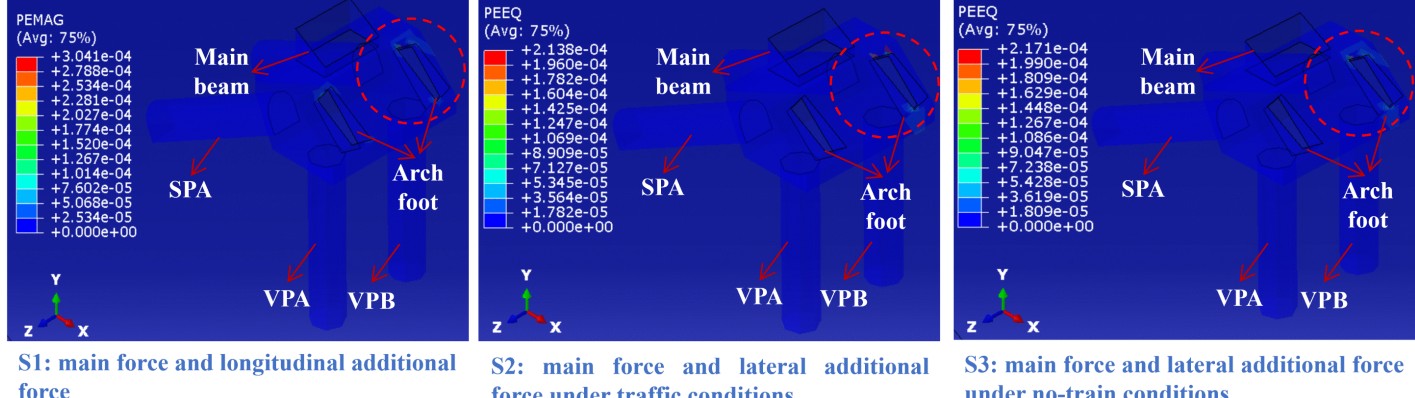

**S1: main force and longitudinal additional force**

**S2: main force and lateral additional force under traffic conditions**

**S3: main force and lateral additional force under no-train conditions**

**Fig 16. Equivalent plastic strain in the soil and rock mass surrounding the arch seat foundation.**

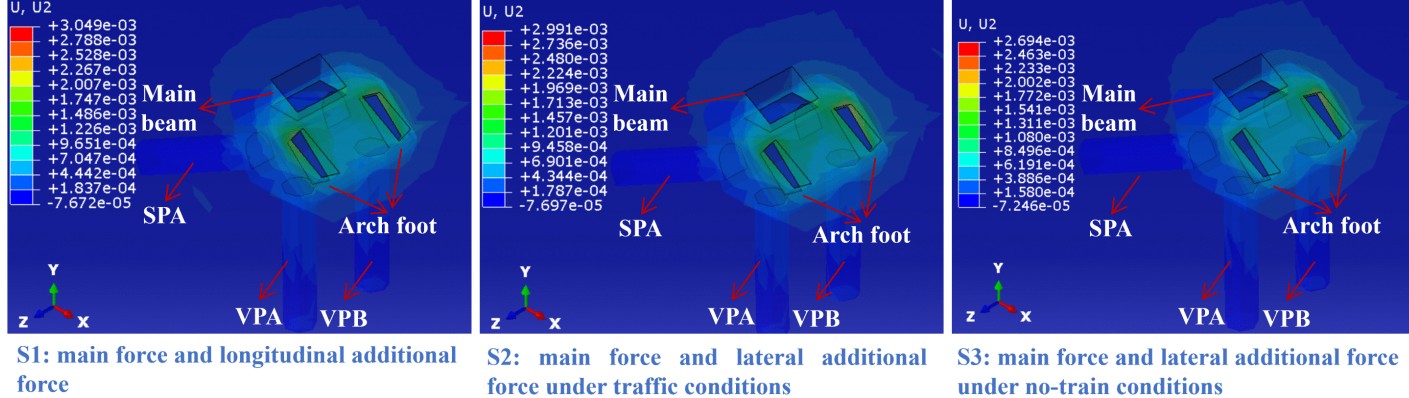

S1: main force and longitudinal additional force

S2: main force and lateral additional force under traffic conditions

S3: main force and lateral additional force under no-train conditions

**Fig 17. Vertical settlement of the soil and rock mass surrounding the arch seat foundation.**

transferred through the arch foot and main beam to the foundation induce localized stress in the surrounding geomaterials, leading to corresponding settlement concentrations. In condition S1, the combined effect of vertical load and braking force amplifies the settlement, with a maximum value of 3.05 mm. Under condition S2, the inclusion of vehicle load and transverse wind action results in a settlement of 2.99 mm, reflecting vehicle-induced stress fluctuations. In condition S3, where only wind load acts in the absence of vehicle load, the settlement decreases to 2.69 mm. The rock slope's anchoring effect along the piles and beneath the cap effectively confines the settlement to the connection area and prevents its propagation to the surrounding pile zone or deeper slope regions. Across all cases, settlement remains localized near the connection interface, with minimal values elsewhere due to boundary constraints. Accordingly, it is essential to monitor settlement accumulation in the connection area, optimize the structural detailing to alleviate stress concentration, and implement ground improvement measures such as grouting to enhance the bearing capacity of the surrounding soil-rock mass. Long-term settlement monitoring is also recommended to ensure the sustained stability of the arch seat foundation.

## 5. Conclusions

The main findings from this study are summarized as follows:

(1) The connection area between the arch seat foundation pile cap and the arch foot represents the primary stress concentration zone. Under condition S2, the peak tensile stress reaches 23.7 MPa, accompanied by the steepest stress gradient. In condition S1, the stress concentration zone is relatively narrower, whereas under condition S3, it is more uniformly distributed. The maximum compressive stress occurs in condition S3. Strict management of complex traffic load combinations corresponding to condition S2 is essential. It is necessary to optimize heavy vehicle routing, monitor transverse temperature gradients, and conduct regular inspections of rebar stress and potential cracking in the connection area to mitigate risks associated with tensile stress concentrations.

(2) The principal stress in vertical piles follows a pattern of high concentration at the pile head, decaying along the shaft. In condition S1, the maximum principal stress at the head of vertical pile VPA reaches 2.74 MPa; in condition S2, the peak decreases to 2.57 MPa; and in condition S3, it further declines to 2.26 MPa. The axial force distribution shows tension at the pile head transitioning to compression at the tip. Under condition S2, the maximum tensile force at the head of vertical pile VPB is approximately 24.24 MN, while in condition S3, the maximum compressive force at the tip of VPA reaches about 26.93 MN. The maximum bending moment, approximately 40 MN·m, occurs at the vertical pile-cap connection. Therefore, countermeasures should prioritize reinforcement at the pile head and long-term monitoring

of stresses and bending moments at the tip of VPA and the head of VPB. The anchoring effect of the rock slope should be utilized to enhance reinforcement at the pile base and mitigate cumulative load-induced damage.

(3) The axial force in slanted piles is characterized by compressive concentration at the pile head. Under condition S1, the maximum compressive force at the head of slanted pile SPA reaches 16.97 MN. The bending moment decays from the pile head, with a maximum value of about 5.8 MN·m, showing minimal sensitivity to load combinations. The distribution on both sides remains highly consistent. Subsequent monitoring should focus on the tensile stress concentration at the head of slanted pile SPA under condition S1 and the stress evolution of slanted pile SPB in condition S3. Moreover, geotechnical anchorage in the inclined segment should be improved to ensure stable collaborative load-bearing performance with the arch seat foundation.

(4) Plastic strain and vertical settlement in the surrounding soil-rock mass are predominantly concentrated in the arch foot-pile cap connection area. Both plastic strain and settlement are most pronounced under condition S1, with a maximum settlement of 3.05 mm. Under conditions S2 and S3, influenced by vehicle and wind loads, the deformation gradient decreases. It is recommended to implement grouting reinforcement in the soil-rock mass of the connection area, optimize the connection structure to alleviate stress concentration, and maintain continuous deformation monitoring. Long-term foundation stability should be ensured by leveraging the anchoring constraints provided by the rock slope.

## Supporting information

**S1 Data. Dataset.**
(RAR)

## Author contributions

**Conceptualization:** Dong Xia.

**Data curation:** Dingxin Zhang, Tong Luo.

**Investigation:** Dong Xia, Guanguo Liu, Dingxin Zhang.

**Methodology:** Dong Xia, Guanguo Liu.

**Project administration:** Dong Xia.

**Resources:** Dong Xia.

**Software:** Dong Xia, Guanguo Liu, Dingxin Zhang, Tong Luo.

**Validation:** Dong Xia.

**Writing – original draft:** Dong Xia, Guanguo Liu.

**Writing – review & editing:** Guanguo Liu, Dingxin Zhang, Tong Luo.

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
