## [Decision Letter · Decision Letter 0]

20 Oct 2025

Dear Dr. Liu,

Thank you for submitting your manuscript to PLOS ONE. After careful consideration, we feel that it has merit but does not fully meet PLOS ONE’s publication criteria as it currently stands. Therefore, we invite you to submit a revised version of the manuscript that addresses the points raised during the review process.

You will see from the referees' comments that additional information needs to be provided, and we ask that this be provided, before we consider you manuscript further.

We look forward to receiving your revised manuscript.

Kind regards,

Zhenhua Li

Academic Editor

PLOS ONE

Journal Requirements:

This work was supported by the Transportation Science and Technology Program Project of Fujian Provincial (Grant No. ZD202401).

This work was supported by the Transportation Science and Technology Program Project of Fujian Provincial (Grant No. ZD202401).

This work was supported by the Transportation Science and Technology Program Project of Fujian Provincial (Grant No. ZD202401).

5. We note that your Data Availability Statement is currently as follows: All relevant data are within the manuscript and its Supporting Information files.

Additional Editor Comments:

You will see from the referees' comments that additional information needs to be provided, and we ask that this be provided, before we consider you manuscript further.

Reviewers' comments:

Reviewer's Responses to Questions

**Comments to the Author**

1. Is the manuscript technically sound, and do the data support the conclusions?

Reviewer #1: Yes

Reviewer #2: Yes

2. Has the statistical analysis been performed appropriately and rigorously?

Reviewer #1: N/A

Reviewer #2: Yes

3. Have the authors made all data underlying the findings in their manuscript fully available?

Reviewer #1: Yes

Reviewer #2: Yes

4. Is the manuscript presented in an intelligible fashion and written in standard English?

Reviewer #1: Yes

Reviewer #2: Yes

Reviewer #1: The paper addresses a practically important topic: the mechanical response of arch-seat foundations of extra-large arch bridges subjected to realistic combinations of traffic, temperature, wind and braking loads. The authors construct a large-scale 3-D finite-element model and examine stresses, axial forces, bending moments, and soil plasticity.

The study is timely, but the current manuscript still needs significant improvement in clarity, completeness of methodology, and discussion of novelty.

1. Significance and Novelty

• The paper is relevant to long-span bridge foundation engineering.

• The use of a full-scale 3-D model with coupled load combinations is worthwhile.

• However, the novelty is not convincingly highlighted. Similar FEM-based investigations of arch-seat or abutment foundations under multi-field loading already exist.

• The manuscript should explain more clearly what is new—for example: the systematic comparison of vertical vs. slanted piles under realistic combined loads, the explicit inclusion of temperature gradients and braking forces, and the link between soil plasticity concentration and foundation design recommendations.

2. Numerical Modeling

Strengths

• Clear description of geometry, boundary conditions, and load cases (S1–S3).

• Sensible mesh-sensitivity study to select an efficient mesh.

Issues & Suggestions

1. Constitutive model:

o The paper uses the extended Drucker–Prager model but does not give the hardening law in full detail (stress–strain curve, yield evolution).

o Justify the choice of friction angle 30°, dilatancy 15°, and whether strain-softening was considered for mudstone.

2. Load application:

o Forces and moments at reference points are tabulated, but the derivation from code-based design loads is not shown; a brief explanation or a supplementary note would improve transparency.

3. Boundary conditions:

o The constraints at the far field of the slope need justification to ensure minimal reflection and realistic stress transmission.

4. Verification:

o The comparison between numerical and theoretical axial-force distribution is useful; more such verification (e.g., deflection of arch foot) would increase confidence.

5. Reproducibility:

o I strongly recommend that the authors provide the Abaqus input file (*.inp) or the CAE file as supplementary material.

This would allow reviewers and future readers to reproduce the results and examine modeling details such as contact definitions, element types, and load steps.

3. Results and Discussion

• The contour plots of stresses and strains are clear, but the captions should define symbols (e.g., VPA, SPA) and load cases for self-containment.

• The physical interpretation is reasonable, but in some places it is descriptive rather than analytical. For example:

o Explain why tensile stresses concentrate at the cap-arch-foot junction (geometry discontinuity, stiffness contrast).

o Discuss potential cracking or fatigue implications of repeated traffic loading.

• The engineering recommendations (steel reinforcement, grouting, monitoring) are sensible but would benefit from a clearer link to the numerical findings.

4. Writing and Presentation

• The manuscript is generally readable but would benefit from language editing to reduce repetition and improve grammar.

• The abstract is somewhat long; consider shortening while keeping motivation, approach, main findings, and practical implications.

• Figures 7–13 are informative; consider including the scale bars and marking critical locations (pile head, pile tip, arch foot).

• Check that all references are complete and that the most recent related work is cited.

5. Specific Comments / Questions

1. How sensitive are the computed stresses to the assumed dilatancy angle and hardening law of the surrounding mudstone?

2. Does the model include any interface/slip between pile and soil, or is it fully bonded?

3. Since traffic loads vary in time, do the authors expect significant cyclic effects on the plastic zone in the arch-foot connection area?

4. Can the authors share the Abaqus input (*.inp) or CAE file as supplementary data to meet open-science and reproducibility standards?

5. Were any field measurements (e.g., strain gauges in piles) available for validation?

6. Recommendation

The paper has the potential to be a valuable contribution, but major revision is required to:

• Emphasize novelty and engineering significance,

• Provide clearer details of the constitutive law and load derivation,

• Add discussion that links numerical findings to physical mechanisms and practical design,

• Supply the Abaqus input or CAE file to enhance reproducibility,

• Improve language and figure presentation.

Reviewer #2: 1. It is suggested to supplement the name of the finite element analysis software used in Section 3.1.

Do both the arch seat foundation and the adjacent slope in Section 3.2 adopt the extended linear Drucker-Prager (DP) model?.It is suggested to supplement the explanation.

2. In Fig. 7, the direction of the MAX value coincides with the legend. It is suggested to distinguish them. In Fig. 8,is the position of the S2 MAX value correct? and supplement the numerical value of S3.

**Do you want your identity to be public for this peer review?** For information about this choice, including consent withdrawal, please see our Privacy Policy

Reviewer #1: No

Reviewer #2: No

---

## [Author Response · Author response to Decision Letter 1]

3 Nov 2025

Detailed responses to the reviewers have been uploaded as a file. The authors sincerely hope that the editors and reviewers understand the confidential nature of the project upon which this research is based; all data must be presented and shared within the confidentiality rules and ethical framework of the relevant authorities. Of course, the authors believe this will not affect the contribution of the paper itself. Thank you for your understanding.

---

## [Decision Letter · Decision Letter 1]

23 Nov 2025

Mechanical response design of arch seat foundation under complex traffic loading

PONE-D-25-51129R1

Dear Dr. Liu,

We’re pleased to inform you that your manuscript has been judged scientifically suitable for publication and will be formally accepted for publication once it meets all outstanding technical requirements.

Kind regards,

Zhenhua Li

Academic Editor

PLOS ONE

Additional Editor Comments (optional):

accept

Reviewers' comments:

Reviewer's Responses to Questions

**Comments to the Author**

Reviewer #1: All comments have been addressed

2. Is the manuscript technically sound, and do the data support the conclusions?

Reviewer #1: Yes

3. Has the statistical analysis been performed appropriately and rigorously?

Reviewer #1: N/A

4. Have the authors made all data underlying the findings in their manuscript fully available?

Reviewer #1: Yes

5. Is the manuscript presented in an intelligible fashion and written in standard English?

Reviewer #1: Yes

Reviewer #1: I am satisfied with the authors' responses on my comments. hence, I am inclined to accept the revised manuscript for publication in the Journal.

**Do you want your identity to be public for this peer review?** For information about this choice, including consent withdrawal, please see our Privacy Policy

Reviewer #1: **Yes: ** Professor Mukhtiar Ali Soomro

---

## [Editor Report · Acceptance letter]

PONE-D-25-51129R1

PLOS One

Dear Dr. Liu,

I'm pleased to inform you that your manuscript has been deemed suitable for publication in PLOS One. Congratulations! Your manuscript is now being handed over to our production team.

Kind regards,

on behalf of

Professor Zhenhua Li

Academic Editor

PLOS One